# Lessons from Learning to Spin "Pens"

**Jun Wang**[*,1]    **Ying Yuan**[*,2]    **Haichuan Che**[*,1]    **Haozhi Qi**[*,3]
**Yi Ma**[3]    **Jitendra Malik**[3]    **Xiaolong Wang**[1]
[1]UC San Diego    [2] Carnegie Mellon University    [3] UC Berkeley
https://penspin.github.io/

Continuous Spinning

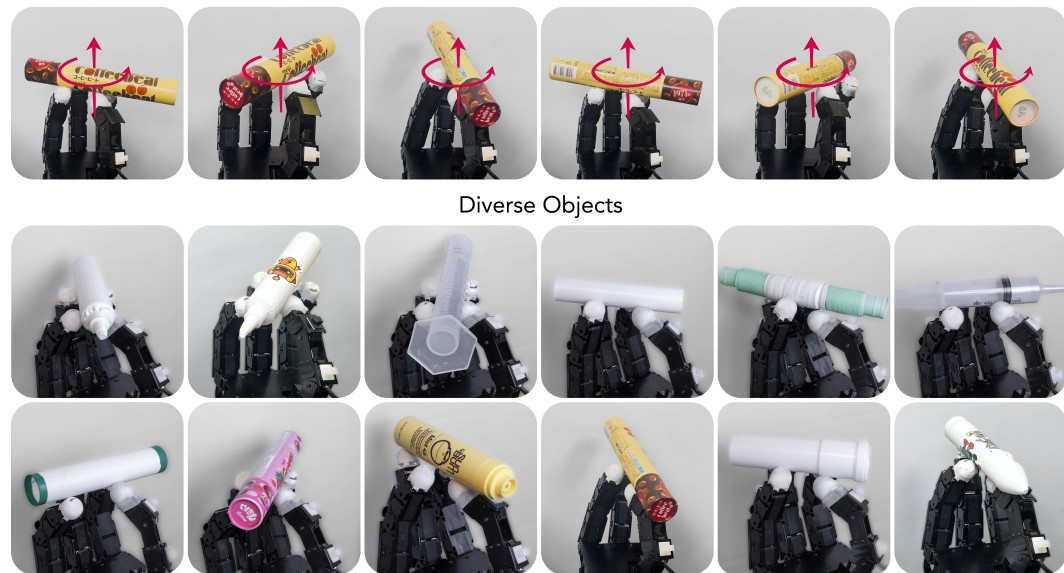

Diverse Objects

Figure 1: **Top row**: Continuous rotation of a pen-like object in hand. **Bottom rows**: Our policy can generalize to a diverse set of pen-like objects with different physical properties, using only proprioception as feedback. More videos are available on our project website.

**Abstract:** In-hand manipulation of pen-like objects is an important skill in our daily lives, as many tools such as hammers and screwdrivers are similarly shaped. However, current learning-based methods struggle with this task due to a lack of high-quality demonstrations and the significant gap between simulation and the real world. In this work, we push the boundaries of learning-based in-hand manipulation systems by demonstrating the capability to spin pen-like objects. We first use reinforcement learning to train an oracle policy with privileged information and generate a high-fidelity trajectory dataset in simulation. This serves two purposes: 1) pre-training a sensorimotor policy in simulation; 2) conducting open-loop trajectory replay in the real world. We then fine-tune the sensorimotor policy using these real-world trajectories to adapt it to the real world dynamics. With less than 50 trajectories, our policy learns to rotate more than ten pen-like objects with different physical properties for multiple revolutions. We present a comprehensive analysis of our design choices and share the lessons learned during development.
**Keywords:** Dexterous In-Hand Manipulation, Pen Spinning, Sim-to-Real

## 1 Introduction

Dexterous in-hand manipulation is a foundational skill for various downstream manipulation tasks. For example, one often needs to reorient a tool in hand before using it. Despite decades of active research in this area [1, 2, 3, 4], in-hand manipulation remains a significant challenge. Manipulating pen-like objects, in particular, is considered one of the most challenging and crucial tasks [5, 6].

---

[*]Equal Contribution.

8th Conference on Robot Learning (CoRL 2024), Munich, Germany.

This capability is highly practical, as many tools, such as hammers and screwdrivers, have similar shapes. Moreover, spinning pen-like objects requires dynamic balancing and sophisticated finger coordination, making it an ideal testbed for advancing dexterous manipulation systems.

Pen spinning has been studied from several perspectives. Classic robotics works demonstrate rotating wooden blocks with open-loop force control [1]. With high-speed cameras and advanced hardware, agile pen spinning can also be achieved [6]. However, these methods rely on accurate object models and cannot generalize to unseen objects. On the other hand, learning-based methods hold the promise of being generalizable with large-scale data. They have indeed achieved significant progress either with imitation learning [7, 8, 9] or sim-to-real [3, 10, 11, 12]. However, they have only demonstrated manipulation of regular spherical or cuboid-shaped objects, and none can extend the capability to pen-like objects. We attribute this to two reasons: For teleoperation-imitation pipeline, current teleoperation systems fail at collecting complex and dynamic demonstrations; for sim-to-real, bridging the gap for dynamic tasks becomes substantially difficult.

In this work, we push the boundaries of learning-based in-hand manipulation systems by demonstrating their capability to spin pen-like objects. Similar to previous approaches [10, 11, 13], we first learn an oracle policy with privileged information using reinforcement learning in simulation. However, when attempting to distill it into a sensorimotor policy, we find the sim-to-real gap too large. While this gap generally exists in previous in-hand manipulation tasks [11, 13], the extreme difficulty of spinning pen-like objects exposes the gap even further. Fine-tuning the policy with real-world trajectories can be one way to mitigate this gap, but it is challenging to collect demonstrations via teleoperation for this dynamic task. Inspired by recent analysis on open-loop controllers [14, 15], we instead collect a high-fidelity trajectory dataset in *simulation* and use it as an *open-loop controller* on the real robot. The successful trajectories in the real world serve as our *high-quality demonstrations*. We then *bridge the sim-to-real gap* by fine-tuning our sensorimotor policy with these real-world trajectories. With simulation pre-training, our sensorimotor policy has the motion prior from diverse data and can adapt to real-world physics with fewer than 50 trajectories.

We conduct comprehensive experiments in both simulation and the real world. In simulation, we identify the key factors that enable the oracle policy to learn the challenging pen-spinning task and generate realistic trajectories. We then evaluate different methods of obtaining a deployable policy in the real world. We also conduct ablation experiments showing the importance of pre-training in simulation. We demonstrate that our policy can adapt to real-world physics with fewer than 50 real-world trajectories. To the best of our knowledge, this is the first learning-based system to achieve continuous spinning of pen-like objects in the real world.

## 2 Related Work

**Classic in-hand manipulation.** In-hand manipulation has been studied for decades [2, 16]. Classical methods rely on an accurate model and analytically plan a sequence of motions to control the object. For example, Han and Trinkle [17] manipulate objects using sliding, rolling, and finger gaiting motions, while Bai and Liu [18] studies the collaboration of fingers and the palm. Mordatch et al. [19] demonstrates object rotation in simulation by trajectories optimization. Li et al. [20] learns a object-level impedance controller for both grasping and rotating objects. Open-loop manipulation also shows surprising robustness and dexterous behavior [14, 15]. Sieler and Brock [21] uses linearized feedback-control for in-hand manipulation with a soft hand. State-of-the-art systems in this category include full $SO(3)$ reorientation using a compliance-enabled hand [22] and an accurate pose tracker [23]. However, most methods cannot manipulate pen-like objects due to their complex and dynamic nature. Extrinsic dexterity [24] can also be used to achieve dynamic manipulation, but a precise model is necessary. In contrast, our method uses human priors to build a simulator environment but does not rely on an accurate model during deployment.

**Learning-based dexterous manipulation.** Learning-based methods make fewer assumptions and hold the promise of being more generalizable as we acquire more data. Recently, significant progress has been made in this field [3, 4]. The advancement mainly comes from two sources: 1) low-cost and accessible teleoperation systems [7, 8, 25, 26, 27, 28, 29, 30] combined with imitation

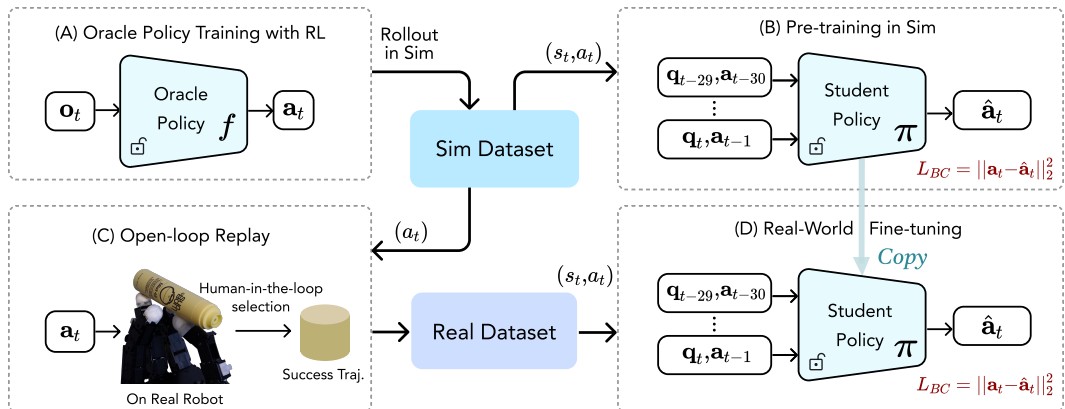

Figure 2: **An overview of our approach.** We first train an oracle policy in simulation using reinforcement learning. This policy provides high-quality trajectory and action datasets. We use this dataset to train a student policy and as an open-loop controller in the real world to collect successful real-world trajectories. Finally, we fine-tune the student policy using this real-world dataset.

learning [31, 32]; and 2) reinforcement learning in simulation [33, 34] combined with sim-to-real [10, 11, 12, 34]. However, both methods have limitations. Current teleoperation cannot support agile and dynamic tasks such as spinning pens, due to the non-negligible communication latency and retargeting errors. On the other hand, sim-to-real approaches demonstrate great generalization and robustness by training policies in randomized environments. They show success in multiple fields such as in-hand manipulation [13, 35, 36, 37, 38, 39, 40, 41], grasping [42, 43, 44], long-horizon tasks [45], and bimanual dexterity [46, 47]. However, the gap between simulation and reality is quite large, and some results are only limited to simulation [48, 49, 50]. Our paper distinguishes itself from all previous work by leveraging the advantages of both fields. We use reinforcement learning in simulation to obtain high-quality demonstrations and use real-world trajectories to bridge the sim-to-real gap.

Our work is also related to several recent works on combining real-world and simulation data. Torne et al. [51] and Wang et al. [52] augment real-world human demonstrations by creating a simulated environment, showing this is helpful for policy robustness. Jiang et al. [53] demonstrates that sim-to-real policies can adapt to real-world complex dynamics with only a few human demonstrations. Our approach also adapts policies trained in simulation to the real world using demonstrations and utilizes simulation data to make the policy more generalizable and robust. However, since our task is more challenging, it is difficult to collect human demonstrations or provide human feedback. Therefore, we need to generate high-fidelity trajectories by learning a policy in simulation.

**Pen spinning.** The specific problem of pen spinning has also been studied extensively due to its challenging nature and practical implications in the real world. Fearing [1] shows an open-loop force control strategy can achieve robust finger gaiting for manipulating a long wooden block. Ishihara et al. [6] and Nakatani and Yamakawa [5] demonstrate high-speed pen spinning using a high-speed robot hand and camera. In the machine learning community, Charlesworth and Montana [54] demonstrate promising results with RL and trajectory optimization. Ma et al. [55] uses a language model for reward design. However, the results are limited to simulation. Bringing simulation results to the real world is a substantially harder task. There are works that involve learning to manipulate long objects using real-world reinforcement learning [56] or augmented with imitation [57], but it can only do less than half a circle and no finger gaiting. In contrast, we achieve continuous pen spinning using a learning-based approach and commercially available hardware.

## 3 Learning to Spin Pens

An overview of our method is shown in Figure 2. Our method consists of three steps. First, we train an oracle policy with privileged information to generate realistic trajectories in simulation. With these trajectories, we pre-train a sensorimotor policy in simulation. We then use these trajectories as

an open-loop controller to generate demonstrations in the real world, which is used to fine-tune the sensorimotor policy to adapt it to the real-world dynamics.

## 3.1 Oracle Policy Training

Obtaining high-quality data for pen spinning is itself a challenging task due to the dynamic and complex movements involved. The current teleoperation system is not suitable due to the non-negligible latency and imperfect retargeting error between the human hand and the robot hand. Alternatively, previous work shows that reinforcement learning can synthesize complex behaviors in simulation [54, 55]. These methods achieve fast and dynamic behavior but may violate real-world physics and hardware constraints. In contrast, we design our approach to generate high-quality trajectories that are realistic enough for use as an open-loop controller in the real world. This is achieved by properly designing the input space, reward function, and initial state distributions.

**Observations.** The observation $o_t$ of the oracle policy $f$ is a combination of the following quantities: joint positions $q_t$, previous joint position target $a_{t-1}$, binary tactile signals $c_t$, fingertip positions $p_t$, the pen's current pose and angular velocity $w_t$, and a point cloud of the pen at the current state $\in \mathbb{R}^{100 \times 3}$. To obtain fine-grained tactile responses, we augment the sensor arrangement in [12] to include five binary sensors on each fingertip (see Figure 5). The point cloud is obtained by transforming points on the original mesh based on the current ground-truth object pose. We encode the point cloud using PointNet [58] as in [35, 59, 60]. We stack three historical states of joint positions and targets as inputs. We also include physical properties such as mass, center of mass, coefficient of friction, and object size in the input [11]. The dimensions of the inputs are detailed in the appendix.

**Actions.** At each step, the action provided by the policy network $f(o_t)$ is a relative target position. The position command $a_t = \eta f(o_t) + a_{t-1}$, where $\eta$ is the action scale, is sent to the robot and it will be converted to torque via a low-level PD controller.

**Reward.** The goal of the policy is to continuously rotate the pen around the $z$-axis. Our reward is defined as a combination of rotation reward and a few energy penalty terms. The reward and penalty terms follow [11, 12]. However, stable gaits do not emerge solely from this. Motivated by [54], we propose another reward $r_z$, a penalty regarding the height difference between the highest and the lowest points on the pen, encouraging the robot hand to keep the pen horizontal during rotation.

In summary, our reward function is ($t$ omitted for simplicity): $r = r_{\text{rot}} + \lambda_z r_z + \lambda_{\text{energy}} r_{\text{energy}}$, where $r_{\text{rot}}$ rewards the pen's rotation velocity and $r_{\text{energy}}$ penalizes the object's linear velocity, deviation from initial joint positions, mechanical work, and torque applied (see appendix for details).

**Initial state design.** Our task fundamentally differs from previous work where the object is placed on the palm [12, 13], a table [10], or fingertip by gravity [11], where there is natural support in those cases. Therefore, using randomly sampled poses does not provide meaningful exploration in our case. We find that a proper design of the initial state distribution is critical for policy training. Designing initial states for

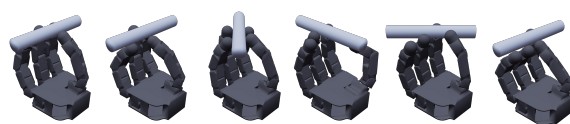

Figure 3: **Visualization of canonical grasp.** Inspired by how humans spin pens, we design six canonical initial poses used to reset the episode. These poses are keyframes where each finger breaks and re-establishes contact.

pen rotation is non-trivial because the initial grasp should be stable enough to facilitate learning subsequent steps of motion. Moreover, exploration can be slow if we repeatedly use the same initial state upon reset. Thus, inspired by human behavior, we manually design multiple patterns of grasping that may occur in the cycle of pen rotation (visualized in Figure 3), and then add noise to generate and filter for a set of stable initial states.

**Policy optimization.** We use proximal policy optimization (PPO) [61] to train the oracle policy. Given the state information, we use a Multi-Layer Perceptron (MLP) for both the policy and value networks. We apply domain randomization to perception inputs, physical parameters, object prop-

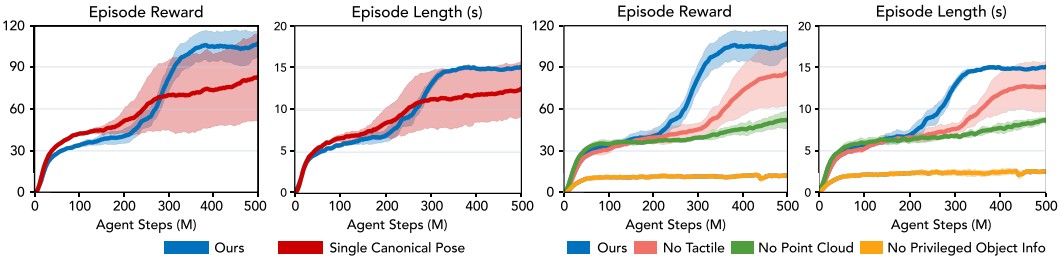

Figure 4: **Learning curves for our policy and different baselines. Left:** Using a well-designed initial distribution is critical. Our method samples the initial states from six proposed canonical states with noise, while *Single Canonical Pose* only samples near one canonical grasp. This has unstable training performance and the finger gaiting does not emerge (see Figure 7 C in appendix). **Right:** The necessity of using visuotactile information and privileged information during *oracle policy* training. We train each policy with 3 seeds.

erties, etc. An episode terminates when reset conditions are met or the agent reaches the maximum number of steps $T$. We prune unnecessary explorations when the pen falls below a height threshold.

## 3.2 Sensorimotor Policy Pre-training

The oracle policy mentioned above can learn smooth and dynamic behavior during simulation training. However, it cannot be deployed because it requires privileged information as input, which is not accessible in the real world. Previous works typically distill the oracle policy into a sensorimotor policy using DAgger [62]. However, we find this approach does not work well for our pen-spinning task. We experimented with either proprioception [11] or adding visuotactile feedback [13, 35]. While the policy with visuotactile feedback can learn reasonable behavior in simulation, the mismatch between simulation and reality is too large for these two modalities. On the other hand, proprioceptive feedback is the most similar and reliable sensing method between simulation and the real world, but the proprioceptive policy cannot converge even in simulation and always drops the object in the first few steps.

For this reason, we propose an alternative approach: we roll out the oracle policy $f$ in simulation, in contrast to previous work using DAgger and rolling out the sensorimotor policy [11, 35], and collect a dataset of proprioception and actions $(s_t, a_t)$. This dataset is used to pre-train a proprioceptive policy in simulation. The goal of this step is to expose the sensorimotor policy to diverse training data. Although training with such data cannot enable direct transfer to the real world due to inaccurate dynamics, it can provide a motion prior, allowing the policy to be efficiently fine-tuned with real-world trajectories.

Following [11], our proprioceptive policy takes 30 steps of joint positions $q_{t-29:t}$ and previous joint targets $a_{t-30:t-1}$ as input. We use a temporal transformer similar to the one used in [35] to model sequential features and an MLP for the policy network. Such pre-training allows our proprioceptive policy to experience a wider range of circumstances, preventing overfitting to specific trajectories.

## 3.3 Fine-tuning Sensorimotor Policy with Oracle Replay

Due to the large sim-to-real gap of our task, we choose to use real-world trajectories to fine-tune the pre-trained sensorimotor policy to adapt to real-world dynamics. However, obtaining real-world trajectories is challenging. Our key observation is that although the oracle policy cannot be directly distilled and zero-shot transferred to the real world, it does provide motion sequences that are difficult to generate using teleoperation. Inspired by recent work that highlights the effectiveness of open-loop controllers for in-hand manipulation [14, 15], we use the trajectories generated by the oracle policy as an open-loop controller in the real world.

Specifically, after training the oracle policy $f$, we test it in the simulation environment with different initial poses. We select 15 trajectories from different initial poses that last longer than 800 timesteps. We record these actions and replay them on the real robot with three training objects (Figure 6). For each replay, we randomly select one of the 15 trajectories. If this open-loop controller can rotate

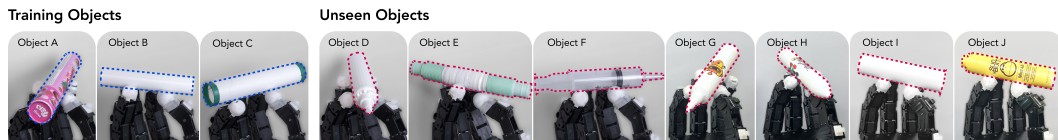

Figure 6: **Training/Test Split of Objects.** We use three training objects to collect real-world trajectories. We evaluate our policy and baselines on both training objects and unseen objects.

| | Training Objects | | | | | | Unseen Objects | | | | | | | | | | | | | |
|---|---|---|---|---|---|---|---|---|---|---|---|---|---|---|---|---|---|---|---|---|
| Object | Object A | | Object B | | Object C | | Object D | | Object E | | Object F | | Object G | | Object H | | Object I | | Object J | |
| Metric | RR.↑ | Suc.↑ | RR.↑ | Suc.↑ | RR.↑ | Suc.↑ | RR.↑ | Suc.↑ | RR.↑ | Suc.↑ | RR.↑ | Suc.↑ | RR.↑ | Suc.↑ | RR.↑ | Suc.↑ | RR.↑ | Suc.↑ | RR.↑ | Suc.↑ |
| Replay | 2.80 | 37.62 | 3.37 | 54.29 | 2.65 | 29.52 | 3.83 | 78.21 | 3.44 | 67.09 | 2.47 | 51.49 | 2.93 | 44.35 | 3.53 | 41.51 | 2.65 | 30.99 | 2.56 | 34.38 |
| P. Distill | N.A. | | | | | | N.A. | | | | | | | | | | | | | |
| V. Distill | 1.85 | 17.65 | 1.57 | 0.00 | 1.70 | 8.33 | 1.57 | 0.00 | 1.57 | 0.00 | 1.57 | 0.00 | 1.57 | 0.00 | 1.57 | 0.00 | 1.57 | 0.00 | 1.57 | 0.00 |
| Ours | 3.43 | 54.93 | 3.38 | 70.00 | 3.62 | 57.55 | 4.10 | 80.65 | 3.50 | 68.18 | 2.71 | 53.33 | 4.47 | 78.02 | 4.63 | 75.79 | 3.64 | 46.60 | 3.49 | 60.47 |

Table 1: **Comparison with different deployable systems.** Oracle Replay achieves reasonable performance but is still inferior to ours. Distillation to proprioceptive policy (P. Distill) fails to converge even during simulation training. Distillation to vision policy (V. Distill) suffers from a significant sim-to-real gap. Many entries are recorded as 1.57 for Vision Distillation due to a consistent failure mode: the thumb and index finger can rotate the object by 90 degrees, but then the object drops. Our method achieves the best performance.

objects more than $2\pi$ in this trial, we store this trajectory in the dataset. We repeat this process until we collect 15 trajectories per object (45 trajectories in total).

Using the learned policy to generate such trajectories has two benefits: First, it naturally provides smoothness driven by our reward definition; Second, compared to alternative approaches such as learning from human videos, it provides trajectory data with actions. We use this dataset to fine-tune our proprioceptive policy $\pi$ to make it adapt to real-world dynamics. Because the proprioceptive policy has already been pre-trained in diverse simulation environments, it can adapt to the real world with fewer than 50 trajectories.

# 4 Experiments

In this section, we compare our approach for pen spinning to several baselines in both simulation and the real world. Specifically, we study 1) the critical design choices in obtaining an oracle policy that can be replayed in the real world; 2) various techniques for sim-to-real deployment.

## 4.1 Experiment Setup

**Object dataset.** In simulation, we only use cylindrical objects with randomized physical properties. During real-world behavior cloning training and testing, we use 3 objects to collect demonstrations and for training, and 7 different objects for evaluation.

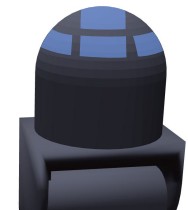

**Evaluation metrics.** In our simulation experiments, we evaluate the Cumulative Rotation Reward and Duration (seconds) [11, 12, 13]. In the real world, we measure the radians of rotation (RR.) over the $z$-axis and the success rate (Suc.). We define success as the rate at which the policy can rotate

Figure 5: Touch sensor (blue) arrangement.

target objects at least 180 degrees, which typically corresponds to the policy completing one circle of finger gaiting, where each finger completes a break and re-establishes contact.

## 4.2 Oracle Policy Training

The goal of the oracle policy is to generate realistic trajectories that can be used both for pre-training the student policy and serving as an open-loop controller in the real world. We compare several critical factors in achieving this, specifically: 1) without a well-designed initial pose distribution; 2) without privileged information; 3) without $r_z$.

**Q1: How does the initial state distribution help policy training?** We study the effect of a well-designed initial state distribution. The results are shown in Figure 4 left. *Single Canonical Pose*

| Object | Training Objects | | | | | | Unseen Objects | | | | | | | | | | | | | | |
|---|---|---|---|---|---|---|---|---|---|---|---|---|---|---|---|---|---|---|---|---|---|
| | Object A | | Object B | | Object C | | Object D | | Object E | | Object F | | Object G | | Object H | | Object I | | Object J | |
| Metric | RR.↑ | Suc.↑ | RR.↑ | Suc.↑ | RR.↑ | Suc.↑ | RR.↑ | Suc.↑ | RR.↑ | Suc.↑ | RR.↑ | Suc.↑ | RR.↑ | Suc.↑ | RR.↑ | Suc.↑ | RR.↑ | Suc.↑ | RR.↑ | Suc.↑ |
| Only Pretrain | 1.89 | 15.15 | 2.44 | 44.87 | 1.70 | 8.11 | 1.74 | 6.86 | 2.13 | 29.35 | 1.98 | 21.05 | 2.06 | 19.77 | 2.11 | 21.59 | 2.68 | 54.08 | 2.14 | 22.22 |
| No Pretrain | 2.62 | 53.66 | 2.34 | 36.84 | 2.29 | 30.00 | 1.92 | 16.53 | 1.88 | 19.61 | 1.90 | 16.42 | 2.09 | 23.86 | 2.15 | 24.72 | 2.92 | **63.22** | 2.41 | 33.70 |
| Ours | **3.43** | **54.93** | **3.38** | **70.00** | **3.62** | **57.55** | **4.10** | **80.65** | **3.50** | **68.18** | **2.71** | **53.33** | **4.47** | **78.02** | **4.63** | **75.79** | **3.64** | 46.60 | **3.49** | **60.47** |

Table 2: **The effect of pre-training and fine-tuning** for our method. We show both components are critical for our method. Without pre-training, the policy tends to overfit to the limited amount of real-world trajectories. With only pre-training, the policy does not work well because of the large sim-to-real gap.

| | #Demo | Training Objects | | | | | | Unseen Objects | | | | | |
|---|---|---|---|---|---|---|---|---|---|---|---|---|---|
| | | Object A | | Object B | | Object C | | Object D | | Object E | | Object F | |
| | | RR.↑ | Suc.↑ | RR.↑ | Suc.↑ | RR.↑ | Suc.↑ | RR.↑ | Suc.↑ | RR.↑ | Suc.↑ | RR.↑ | Suc.↑ |
| | 15 | 1.80 | 14.29 | 1.82 | 15.79 | 1.57 | 0.00 | 1.75 | 11.11 | 1.84 | 13.04 | 1.57 | 0.00 |
| No Pretrain | 45 | 2.62 | 53.66 | 2.34 | 36.84 | 2.29 | 30.00 | 1.92 | 16.53 | 1.88 | 19.61 | 1.90 | 16.42 |
| | 75 | 2.93 | **76.67** | 2.78 | 40.00 | 2.57 | 43.33 | 2.36 | 26.67 | 2.09 | 23.33 | 1.96 | 15.00 |
| Ours | 45 | **3.43** | 54.93 | **3.38** | **70.00** | **3.62** | **57.55** | **4.10** | **80.65** | **3.50** | **68.18** | **2.71** | **53.33** |

Table 3: **We study whether having more demonstrations could substitute for simulation pre-training.** We find that although the No Pretrain baseline improves as we increase the number of demonstrations from 15 to 75, it still performs worse than our method, especially on unseen objects. This indicates that training with much more diverse data in simulation is beneficial and can also avoid overfitting to certain objects.

samples states around one canonical hand pose, as used in [11, 35]. In contrast, our method defines multiple canonical hand poses inspired by how humans spin pens and achieves better performance compared to using a single canonical pose. We also emphasize that although the curve for single canonical init does increase over time, the finger gaiting cannot emerge, and this policy cannot escape from the local minima. We visualize the behavior in Figure 7 (c) in our appendix and find the finger does not break contact with the object and fails to achieve more than one revolution.

**Q2: How does privileged information help policy training?** We study the importance of privileged information in Figure 4 right. Unlike [11], the oracle policy cannot be trained only with simple object properties such as object position. We find that without tactile feedback or a point cloud, the policy does not achieve good enough performance. The shape of the pen is important as the policy needs to know when to lift the fingers to spin the pen. Privileged information such as the object's physical properties and finger positions is also critical, without which the policy does not converge.

**Q3: How does z-reward help policy training?** We study the effect of z-reward $r_z$, shown in Figure 7 (b) in our appendix. Although the trajectories look similar to our approach at first glance, the object gets tilted at configurations in the third and sixth sub-figure. This can barely succeed in real-world replay. In contrast, policies trained with the z-reward rotate the pen more stably, keeping the pen approximately horizontal, which facilitates data collection in real-world replay.

### 4.3 Sensorimotor Policy Training

Although the oracle policy achieves great performance in simulation, it cannot be directly deployed in the real world. To address this issue, we use it as an open-loop controller to collect real-world trajectories. We also pre-train a proprioceptive policy in simulation and fine-tune it using this dataset. We compare our method with several alternatives in the real world. The results are shown in Table 1.

**Q4: Is oracle replay a good enough controller?** We design our oracle policy so that it achieves decent performance in the real world (Oracle Replay). However, it still performs worse than our method. On Training Objects A/B/C, our method achieves 15%-30% better performance in terms of success rate. On Unseen Objects D/E/F, which are considered out-of-distribution, our method achieves a 10% increase in the radius rotated, despite having a similar success rate. Our method also achieves 15%-30% success rate improvements on objects I/J/K. This result demonstrates that our method generally achieves a longer radius rotated compared to the oracle replay because it is also pre-trained in simulation with more diverse data.

**Q5: Does distillation work for pen spinning?** Previous approaches demonstrate promising results by distilling the oracle policy into the sensorimotor policy [11, 35, 13] using DAgger. However, this approach does not work for our dynamic and contact-rich task (Figure 1). First, we try to use segmented depth [35] or two endpoints of the pen, and the visuotactile policy can achieve reasonable performance in simulation. However, the sim-to-real gap is significantly larger compared to previous works. In our real-world deployment, the objects oscillate a lot, making the image distribution far removed from the training one. Secondly, proprioceptive feedback does not have this problem, but using proprioception alone does not achieve good performance in simulation.

**Q6: How do pre-training and fine-tuning contribute to the final performance?** Our approach is first pre-trained in simulation and then fine-tuned using real-world data. We study the contribution of each part in Table 2. With only pre-training, the policy has limited effectiveness in the real world. It rarely completes finger gaiting on Objects C and D, and the success rate is also low for the remaining objects. This is mainly because the physics gap between simulation and reality becomes more significant in our task. With only behavior cloning, the approach also does not perform well. For this experiment, we use ACT [63] as the architecture, which is one of the best imitation learning algorithms. It has a 50% lower rotation radius on Training Objects. On out-of-distribution objects, the success rate drops to less than 20%, indicating that its generalization capability is limited.

**Q7: Can simulation pre-training be replaced by more demonstrations?** We also study whether increasing the number of real-world demonstrations can substitute for the advantages gained from pre-training in simulation. The results are shown in Table 3. We find that although the performance of the No Pretraining baseline can be improved with more demonstrations, it gradually saturates when increasing the number of demonstrations from 45 to 75. In addition, the major improvements come from the training objects (A/B/C), while the performance on unseen objects (D/E/F) is still far worse compared to our methods. This indicates that solely relying on real-world trajectories is likely to overfit to certain objects.

### 4.4 Qualitative Experiments

In addition to the objects we present in the quantitative study, we also try more different objects for our policy and try to push the limits on objects that are significantly out-of-distribution. Examples are shown in our Figure 1 and our project website.

## 5 Conclusion and Lessons

In this paper, we present the first learning-based approach for spinning pen-like objects. Through our extensive experiments, we share the lessons we learned as follows:

- **Simulation training requires extensive design for exploration**, such as the proper design of initial distributions to aid exploration and using privileged information to facilitate policy learning.
- **Sim-to-Real does not directly work** for such contact-rich and highly dynamic tasks. Even when isolating touch and vision, the pure physics sim-to-real gap remains significant and cannot be bridged by extensive domain randomization alone.
- **Simulation is still useful for exploring skills.** The dynamic skill of spinning pens with a robotic hand is nearly impossible to achieve with human teleoperation and imitation learning alone. Reinforcement learning in simulation is critical for exploring feasible motion.
- **Only a few real-world trajectories are needed for fine-tuning.** Although a proprioceptive policy learned purely in simulation does not work directly in the real world, it can be fine-tuned to adapt to real-world physics using only a few successful trajectories.

**Limitations.** We have identified several key bottlenecks of using vision and touch during sim-to-real for this dynamic task. However, we are not stating they should not be used. Humans do not seem to need vision to spin a pen, but touch feedback seems important. In future work, we will explore whether using them can help further improve performance. Currently, the system is only capable of rotating along $z$-axis, it is also a promising direction to extend it to general multi-axis rotation. In addition, our work assumes the object is placed at a stable grasp position following previous work [11, 10]. Incorporating more advanced grasping work [64] or consider chaining different skills [45] together would make our work more general.

**Acknowledgments**

Xiaolong Wang's lab is supported, in part, by Amazon Research Award, Intel Rising Star Faculty Award, and Qualcomm Innovation Fellowship, and gifts from Meta. Haozhi Qi and Jitendra Malik are supported in part by ONR MURI N0001421-1-2801. Haozhi Qi and Yi Ma are supported in part by ONR N00014-22-1-2102.

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

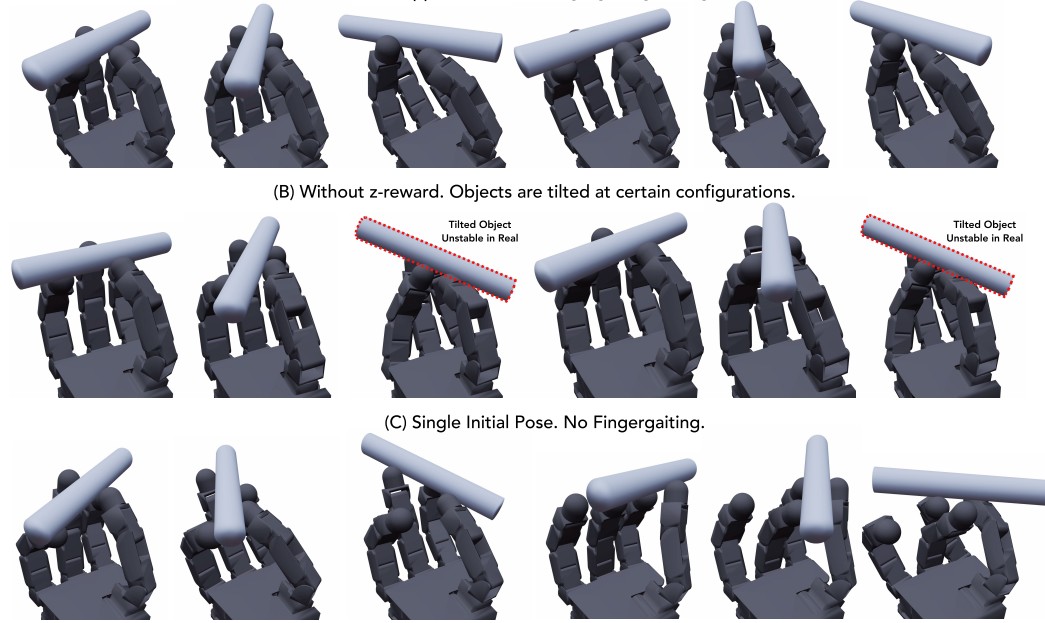

Figure 7: **Importance of $r_z$ and initial state design.** (a) Our policy spins the pen in a smooth and stable manner, with the pen mostly horizontal. (b) Policies trained without the $r_z$ tend to make the pen more tilted during rotation. This behavior is unstable and cannot be used as an open-loop controller in the real world. (c) Initializing with a single canonical state lacks exploration and cannot learn finger gaiting.

# A    Additional Experiments

In Q4 of our experiment section, the initial hand / object configurations are chosen from the initial configurations of the replay trajectory dataset. This setting brings advantage to the oracle replay baseline. To comprehensively study the performance, we conduct new real-world experiments where the objects are initialized at a randomly chosen stable grasp. We choose 10 random grasps, do 5 trials for each. For each random grasp, we generate the oracle replay trajectory by running the oracle policy in the simulator and select the best trajectory among 1000 simulated environments. In this setting, our method achieves 22.0% (from 54% to 78.0%) better success rate on object D, 36% (from 46 to 82) on object E, and 40% (from 34% to 74%) on object F. The reason that our policy does not drop compared to the number if Table 1 is because it has seen much more diverse data via simulation pre-training.

# B    Implementation Details

## B.1    Setup

**Hardware setup.** We use the Allegro Hand for our hardware experiments. The Allegro Hand has four fingers, each with 4 degrees of freedom. Our neural network outputs the joint position target at 20 Hz, which is sent to a low-level PD controller operating at 333 Hz.

**Simulation setup.** We use Isaac Gym [34] for our simulation training. To obtain additional tactile feedback for oracle policy training, we simulate 20 tactile sensors around the fingertips, with 5 on each fingertip. We gather the contact signal from each sensor and binarize the measurement based on a pre-defined threshold [12, 13]. In simulation, the control frequency is 20 Hz and the simulation frequency is 200 Hz.

| Hyper-parameters | Values |
|---|---|
| $\lambda_{\text{rot}}$ | 1.0 |
| $\lambda_{\text{z}}$ | -1.0 |
| $\lambda_{\text{vel}}$ | -0.3 |
| $\lambda_{\text{diff}}$ | -0.1 |
| $\lambda_{\text{ang}}$ | -0.3 |
| $\lambda_{\text{torque}}$ | -0.1 |
| $\lambda_{\text{work}}$ | -1.0 |

Table 4: Hyper-parameters for the reward function.

| Obs Type | Dimension |
|---|---|
| $q_t$ | $\mathbb{R}^{3\times16}$ |
| $a_{t-1}$ | $\mathbb{R}^{3\times16}$ |
| $c_t$ | $\mathbb{R}^{32}$ |
| $p_t$ | $\mathbb{R}^{4\times3}$ |
| $w_t$ | $\mathbb{R}^{7}$ |
| PointCloud | $\mathbb{R}^{100\times3}$ |

Table 5: Dimensions of the inputs of the oracle policy.

| Hyper-parameters | Values |
|---|---|
| # environments | 48 |
| # steps | 512 |
| # minibatches | 4096 |
| # epochs | 2000 |
| learning rate | 1e-3 |

Table 6: Hyper-parameters for training the student policy in the simulation.

## B.2 Training Hyper-parameters

Our reward function is a combination of $r_{\text{rot}}, r_{\text{z}}$ and $r_{\text{energy}}$. The energy reward consists of $r_{\text{vel}}, r_{\text{diff}}, r_{\text{ang}}, r_{\text{torq}}$, and $r_{\text{work}}$. Here, $r_{\text{vel}}$ penalizes the pen's linear velocity, $r_{\text{diff}}$ discourages the hand's pose from deviating much from its initial pose, $r_{\text{ang}}$ penalizes the pen's angular velocity above a pre-defined threshold to encourage stable rotation, $r_{\text{torq}}$ penalizes large torques, and $r_{\text{work}}$ penalizes the work of the controller. We follow the same definition of reward in [35]. We combine the above rewards with weights listed in Table 4.

We detail the dimensions of the inputs of our oracle policy in Table 5. We train our oracle policy with PPO, and the training hyper-parameters are shown in Table 7. Specifically, we train with 8192 parallel environments. Each environment gathers # steps data to train in each epoch of PPO. The data is split into # minibatches and optimized with PPO loss. $\gamma$ and $\lambda$ are used for computing generalized advantage estimate (GAE) returns. We use the Adam optimizer to train PPO and adopt the gradient clip to stabilize training. We train 500 million agent steps in total, which takes less than one day on a single GPU. We train our student policy with Behavior Cloning, and the training hyper-parameters are shown in Table 6. We collect approximately 50M steps of data in total.

| Hyper-parameters | Values |
|---|---|
| # environments | 8192 |
| # steps | 12 |
| # minibatches | 16384 |
| # Agent Steps | 500000000 |
| $\gamma$ | 0.99 |
| $\lambda$ | 0.95 |
| learning rate | 5e-3 |
| clip range | 0.2 |
| entropy coefficient | 0.0 |
| kl threshold | 0.02 |
| max gradient norm | 1.0 |

Table 7: Hyper-parameters for training the oracle policy.

## B.3 Domain Randomization Parameters

The domain randomization parameters are listed in 8.

## B.4 System Identification

We tune the P and D gain in the low level controller according to the following two metrics: 1) Before we do data collection, we first replaying several finger gait trajectory without objects in-hand. We do this for both sim and real and compare the errors between joint positions. We try to

| Parameter | Range |
|---|---|
| Object Scale | x[0.95, 1.05] |
| Mass | [0.01, 0.02] kg |
| Center of Mass | [-0.1, 0.1] cm |
| Coefficient of Friction (obj and fingertip) | [0.3, 3.0] |
| External Disturbance | (0.2, 0.25) |
| PD Controller Stiffness | [2.5, 3.5] |
| PD Controller Damping | [0.09, 0.11] |
| Observation Noise | N(0, 0.02) (rad) |
| Action Noise | N(0, 0.01) (rad) |

Table 8: **Domain Randomization Parameters.** The object scale range is multiplied by the original scale. Observation and action randomizations follow a Gaussian distribution with the specified radius. Other randomizations are uniformly sampled from the specified range. Following [3], we apply a random disturbance force to the object during training whose scale is $0.2m$ with probability $0.25$ where $m$ is the object mass.

minimize the error by tuning the P and D gains simultaneously in sim and real. 2) We also command the sin and cos waves of each joint and observe the errors between sim and real. We also include action noises during training so that the policy can be robust to real-world actuator noises.

