# OpenReview forum: "Lessons from Learning to Spin “Pens”"
_robot-learning.org/CoRL/2024/Conference — CoRL 2024_

### Official Review · Reviewer_U3nL · 2024-07-18
**Interesting results and insights for a long-lasting dexterous manipulation problem**

**Originality:** 4
**Technical Quality:** 3
**Clarity Of Presentation:** 4
**Potential Impact:** 3
**Recommendation:** 4
**Confidence:** 5

**Review:**

The paper targets the challenging problem of spinning a pen in-hand. The main contribution lies in an exploration of many design choices in learning the simulated policy and a strategy of blending the real data that are hard to acquire through teleoperation due to the task dynamic. Many insights on the privileged state and distillation design are discussed. The importance of resetting environment to critical frames in successful execution and terminating failing trials are highlighted. The interesting findings on using proprioception and open loop trajectories from high-quality simulated episodes might be inspirational for addressing other tasks. The real-hardware evaluation looks solid , though some observations are not fully addressed.

Strengths:

- Impressive real robot results on a challenging dexterous manipulation problem.

- Well documented and organised insights on the design consideration, which are mostly reasonable.

- Interesting strategy of augmenting real dynamics data which might inspire other hard-to-demonstrate tasks.

Weakness:

- Limited insights on the design of the real data augmentation strategy.

- Some result presentation can be clearer.

- Minor: the literature review misses the works on rotating objects with trajectory optimisation through contact (a) and model-free compliant control (b).

(a) Mordatch et al., Contact-invariant optimization for hand manipulation, SCA 2012
(b) Li et al., Learning object-level impedance control for robust grasping and dexterous manipulation, ICRA 2014

**Quality Of The Limitations Section:**

3

**Questions For Rebuttal:**

- Any example to show what typical failure modes the real data help to correct? The realized rotating radians on average are still less than $2\pi$, what are blocking a spinning lasting longer?

- How many successful examples we can typically obtain from executing the 15 open-loop trajectories? Are they just the numbers under oracle reply in Table 1?

- Why the oracle replay appears to work better on out-of-distribution objects D/E/F while much poorer for in-distribution objects? Why the improvement from real data augmentation is more significant for in-distribution objects while marginal for D/E/F?

- How large is the discrepancy between in-distribution and out-of-distribution objects in terms of mass distribution, geometry and surface materials?

-  The augmentation seems to rely on a few seeding sim-to-real trials which might be a bit lucky-based. What if we could not obtain sufficient successful real trials?

- Would more exhaustive domain randomization increase the chance of getting successful real replays or even lead to zero-shot sim2real? What specific physical parameters are randomized here? Why not considering randomizing the actuator parameters or communication delay given the task is time-critical?

- Z-reward design seems vital to pruning out invalid explorations. Will a curriculum learning, e.g. with gradually increased gravity, help on efficient exploration?

- The ultimate hardware control is 20 Hz while both simulated control and engine step operate at 200 Hz. How is the simulated control aligned to the real setup? Is it just receiving proprioception less frequently?

-  Since the simulation is designed to reset at predefined contact modes, will a motion planning algorithm with this predefined constraint switching as a mode-search skeleton be more efficient to generate simulation trajectories?

- The ultimate motion looks conservative: it looks more like carefully supporting the pen with tips and nudging it with tiny motions, instead of with more significant spinning motion with more phalange contacts as humans. Will such a behavior harder to learn given the current simulation and hardware setup?

**Robotics Focus:**

4

**Summary Of Paper:**

The paper explores sim-to-real designs with an augmentation of real world data to solve in-hand spinning pen-shaped objects, with a success rate of 50-80% on the real-hardware and the capability of generalizing to unseen objects.

**Summary Of Recommendation:**

The paper still has room to improve on its analysis about the method design and experimental results. However, a contribution for such a challenging task on real-hardware and novel objects should already place it above the acceptance threshold. Edit: the authors' rebuttal helps to address some of the limitations. The flaws on lacking systematic and well-grounded conclusions persist. However, I believe the paper has made visible contribution that deserves discussion at the conference.

---

### Official Review · Reviewer_defC · 2024-07-19

**Originality:** 2
**Technical Quality:** 3
**Clarity Of Presentation:** 3
**Potential Impact:** 2
**Recommendation:** 3
**Confidence:** 4

**Review:**

Strengths:
1. “Spin a pen with a multi-finger robot hand” is one of the most challenging tasks for dexterous manipulation. Although the task is simplified, seeing the system working in the real world and can work on multiple objects is refreshing.
2. This work introduces a novel teacher-student training pipeline, where the demonstrations are replayed on a real-world robot and the successful trajectories are used to fine-tune the policy learned with simulation data. This idea is neat and brings two key benefits: 1) it bridges the sim-to-real gap by leveraging a few real-world, automatically collected data when zero-shot transfer does not work, and 2) it greatly improves sample efficiency.

Questions:
1. Seems like most pen-shaped objects have similar thickness and are evenly weighted along the entire shape. The objects are assumed to be held in the middle at the beginning of the task. The robot motions seem like a specific type of periodic replay. I hope the authors can provide more details on the differences between the training and novel objects, and explain what truly makes generalization difficult here.
2. In the introduction, the authors mention the importance of “dynamic balancing” the pen-like object during spinning. However, there is no object state information input to the policy, which makes me feel the policy does not learn balancing at all. If that is the case, the application of “pen spinning” in this work would be very similar to (1) but only rotate along the z-axis (and without any sensing). The technical method this work uses to help learn dynamic balancing is unclear to me.
3. What would happen if sim2real does not work in the early stages and all the collected trajectories fail? Is the method based on the assumption that a functioning sim2real policy is achievable at the beginning?
4. This work assumes that the action distribution and the low-level PD controller are the same in both simulation and real-world settings, which is not true in most cases. This might also cause most real-world replays to fail, resulting in no successful trajectories being gathered. I hope the authors can introduce more details about aligning action and controller between simulation and real-world settings, and how to guarantee successful trajectories during open-loop replay.
5. The stable initial state is manually specified, which seems to be a limitation for serving as a general approach in other manipulation tasks. Some prior works like (2) and (3) have studied the relation between the way of grasping and following manipulation tasks. It would be nice to add some discussion in the future work section about potential arm+hand motion with both grasping and spinning. This is also a limitation of this work.

(1). General In-Hand Object Rotation with Vision and Touch

(2). Dexterous functional grasping

(3). Sequential dexterity: Chaining dexterous policies for long-horizon manipulation

**Quality Of The Limitations Section:**

2

**Questions For Rebuttal:**

Same as the limitations mentioned above.

**Robotics Focus:**

4

**Summary Of Paper:**

This work introduces a method for dexterous pen spinning tasks. Prior approaches struggle due to poor human demonstrations and sim-to-real gaps. Using reinforcement learning, the authors train a policy in simulation and generate high-quality trajectory data. This data is used for pre-training and real-world trajectory replay, requiring fewer than 50 real-world examples to fine-tune the policy. The result is a system that can successfully rotate various pen-like objects with different properties.

**Summary Of Recommendation:**

Overall, this work introduces a novel method for sim-to-real dexterous manipulation with good real-world results. However, the assumptions of the controller and task setup need more clarification. Therefore, I start with weak reject and might adjust based on the authors response.

---

### Official Review · Reviewer_rqxA · 2024-07-20
**Good study of pen spinning but limited contribution**

**Originality:** 3
**Technical Quality:** 4
**Clarity Of Presentation:** 4
**Potential Impact:** 2
**Recommendation:** 3
**Confidence:** 4

**Review:**

Strength:

1. The paper is well written and most of the technical details are presented clearly in the main text.

2. The figures are nicely done, especially Fig. 1 and Fig. 4. I pick up the idea very quickly from them.

3. I appreciate the thoroughness of the experiment section, carefully ablating the different components of the pipeline.

Limitation:

1. I think the overall contribution is too limited. The use of distillation and real-world fine-tuning is well studied.

2. The main novelty, to me, lies in using open-loop replay on hardware. But meanwhile, I also find it very surprising that this kind of open-loop action trajectories from simulation can work quite well in real, e.g., 78.21% success rate with Object D (Table 1). To me, this means the task is not that difficult, and the sim-to-real gap is not big. Maybe the authors can clarify this a bit.

3. Also, the overall improvement from the oracle baseline is not very impressive in Table 1. I think with multi-stage training the success rates should aim above 80% for the different objects. if there is some fundamental limit, please clarify.

**Quality Of The Limitations Section:**

2

**Questions For Rebuttal:**

It would be good to mention how the policy is parameterized, e.g., is it a Gaussian distribution, what is the noise scale for the distribution, and whether the noise scale is fixed or learned.

**Robotics Focus:**

4

**Summary Of Paper:**

This paper presents a system study on dexterous hand rotating pens. The overall approach involves oracle simulation training, distillation, open-loop replay in real, and another fine-tuning in real. The different components of the pipeline are ablated in the experiments to showcase their importance.

**Summary Of Recommendation:**

I think the paper does a thorough study on pen spinning using mostly existing methods, but I am concerned with the limited overall contribution.

---

### Official Review · Reviewer_qCCU · 2024-07-20
**Interesting, if cursory, study of a 'grand challenge' in dexterous manipulation**

**Originality:** 4
**Technical Quality:** 4
**Clarity Of Presentation:** 4
**Potential Impact:** 3
**Recommendation:** 3
**Confidence:** 4

**Review:**

**Summary:** The authors consider the problem of “pen spinning,” a long-challenging task in dexterous manipulation. In particular, the authors present a “systems” result, showing how to combine existing results (with solid design decisions) to enable robust hardware performance.

In particular, the authors train a ‘teacher’ policy with privileged information in simulation, run policies open-loop in real world, and then fine-tune on real-world data. They show that the “oracle” simulation policy can be adapted to real-world settings in <50 trajectories, which I regard to be remarkable data efficiency (and impressive hardware results).

**Strengths:**

- Considers an ambitious, contact-rich dexterous manipulation problem.
- Very thorough literature review that provides an excellent perspective on historical and current approaches to dexterous in-hand manipulation.
- Strong hardware results showing the proposed pretraining + fine-tuning approach is plausible for this problem.
- Defines a good problem for dexterous manipulation (pen-spinning), although still relatively quasistatic compared to the “typical” definition of “pen spinning”.

**Limitations:**

- Performance is better than baselines but still underwhelming qualitatively. Work still needs to be done to get the dynamic, robust motion the authors are imagining in their introduction.
- Baseline comparisons could be stronger, especially w/r/t imitation learning. In particular, strong comparisons would be vs. diffusion policies with imitation of teleoperated demonstrations. I acknowledge this approach has numerous severe challenges (dynamic demo collection is extremely difficult with current telop pipelines for dexterous hands) but think this comparison would strengthen the paper and the "story" motivating an RL approach here overall (especially since the experimental results, as the authors also acknowledge, are relatively underwhelming).

-----

**Summary after rebuttal phase:**
I largely stand by my original review (aside from some original confusion, now deleted, re: the RL architecture used here). I think the other reviewers are correct to pick at the heavy use of open-loop replay in hardware, and think the paper suffers a lack of clarity around these topics. Overall, however, I view this paper as mostly a *negative result* / temperature check for how well current techniques can perform on a difficult task. I would argue our community needs more of these results (compared to / in addition to minor advancements on less ambitious tasks) and think this clears the bar for CoRL.

**Quality Of The Limitations Section:**

1

**Questions For Rebuttal:**

- The authors draw a clear contrast between their approach and imitation learning throughout, they do not actually baseline against IL results like diffusion policies. Does the proposed method outperform pure imitation in hardware (which is difficult to collect, but suffers no sim-to-real gap)?
- What are some limitations of the current method? Or, put differently, what would be required to get a robot hand to spin pens dynamically like humans can do?

**Robotics Focus:**

4

**Summary Of Paper:**

Simulation-based pre-training and real-world fine-tuning enable sim-to-real transfer for challenging in-hand manipulation.

**Summary Of Recommendation:**

Weak accept; 'systems' papers are under-appreciated in our community + this is a great study of how exiting techniques work for a 'grand challenge' in manipulation.

---

### Author Rebuttal · Authors · 2024-08-12

We thank all reviewers for the helpful feedback. We attach an updated version of our paper below.

---

### Decision · Program_Chairs · 2024-09-04

**Decision:**

Accept

**Comment:**

The submission analyzes sim2real transfer for manipulating pen like objects.

The submission tackles an ambitious, contact-rich dexterous manipulation problem and includes a thorough literature review. It shows real robot results and provides detailed analysis. It proposes a novel strategy of augmenting dynamics data.

However, insights on the design of the real data augmentation strategy still need to be improved and some result presentations could be clearer. Overall, the performance, while better than baselines, is still underwhelming and stronger baseline comparisons would be helpful. A key point lies in using open-loop replay on hardware, which might not work for other tasks and could be more broadly explored. Partial simplification arises from the stable initial state being manually specified.

The paper has improved during the rebuttal via detailed communication between reviewers and authors even though not all points have been addressed there has been a shift in reviewer evaluation. Based on the above the paper is recommended for acceptance. Please follow up on the remaining open points for a potential camera ready version.